# The Impact of Integrated Harvesting Systems on Productivity, Costs, and Amount of Logging Residue in the Clear-Cutting of a *Larix kaempferi* (Lamb.) Carr. Stand

Heesung Woo [1,†] , Eunjai Lee [2,†] , Mauricio Acuna [3] , Hyunmin Cho [4] and Sang-Kyun Han [5,*]

1    College of Forest and Environmental Sciences, Kangwon National University, Chuncheon 24341, Republic of Korea
2    Forest Technology and Management Research Center, National Institute of Forest Science, Seoul 02455, Republic of Korea
3    Forest Industries Research Centre, University of the Sunchine Coast, Locked Bag 4, Maroochydore, QLD 4558, Australia
4    Department of Forestry Management, Kangwon National University, Chuncheon 24341, Republic of Korea
5    Division of Forest Science, Kangwon National University, Chuncheon 24341, Republic of Korea
*    Correspondence: hsk@kangwon.ac.kr
†    These authors contributed equally to this work.

**Abstract:** Two integrated harvesting methods have been primarily applied to increase the opportunity for forest biomass utilization. In Korea, small shovels with a carrier for cut-to-length harvesting (CTL system) and tower yarders for whole-tree harvesting (WT system) are commonly used for the transportation of tree assortments (i.e., sawlogs and logging residue). No previous studies are available in South Korea that have compared and highlighted the operational performance and yield of logging residues between the CTL and WT harvesting systems. Thus, our study's main objectives were to (1) evaluate the productivity and costs of the two harvesting systems through a standard time study method and (2) estimate the amount of harvesting logging residue at the landing. The productivities of the CTL and WT systems were 1.45 and 2.99 oven-dried tons (odt)/productive machine hour (PMH), at a cost of 86.81 and 45.41 USD/odt, respectively. In the WT system, the amount of logging residue (2.1 odt/ha) collected at the landing was approximately four-times larger than that of the CTL system (0.5 odt/ha). Our results suggested that the WT system is a less expensive and more suitable system when there are markets demanding logs and biomass, whereas the CTL system remains a less expensive option for stem-only harvesting. Furthermore, these results are important for estimating the economic and environmental amount of residue that could be potentially recovered and utilized from the forest types included in the study.

**Keywords:** biomass recovery operation; forest residue; cut-to-length; whole tree harvesting; system balance





## 1. Introduction

South Korea (hereafter Korea) is the eighth-largest energy consumer in 2020 and imports almost 94% of its total energy and resources, including liquefied natural gas, coal, and crude oil [1]. However, Korea's energy policy is progressing and fostering the use of renewable energy to reduce greenhouse gases to zero and reach carbon neutrality by 2050 [2]. Since 2012, the Renewable Portfolio Standard has been the primary policy instrument for promoting renewable energy [3]. In 2019, bioenergy produced approximately 4.2 million tons of oil equivalent (toe), representing 40% of Korea's total renewable energy production (10.3 million tons; [4]). Wood-based energy (1.5 million tons) was the dominant renewable energy source, representing 37% of Korea's total bioenergy production [4].

Wood-based energy consumes various types of fuel products made of woody biomass such as bio-solid refuse fuel (SRF), wood chips, and wood pellets. The primary ingredient

used to produce wood chips and the wood pellets, which are dominant fuel products substituting bituminous coal at power plants, is forest residue. Forest residue includes non-merchantable trees, small-diameter trees, tops, limbs, and leaves that come from the forest after merchantable-timber harvesting. Therefore, with an increased interest in forest-residue utilization as a wood-based energy resource, efficient forest operation management planning is required to optimize the productivity of biomass-recovery forest operations. To do this, integrated harvesting has emerged as an attractive alternative for sustainable renewable energy production and for the economic viability of biomass harvesting operations.

Globally, integrated harvesting is becoming a primary alternative for forest-fire risk and climate change mitigation, as well as improved forest resource utilization and cost-efficiency—particularly in Europe and North America [5,6]. Various integrated harvesting systems have been applied to implement biomass-harvesting technologies (e.g., single-pass and double-pass operations) under a wide range of harvest unit conditions, such as slope, road surface, and landing size [5,7,8]. Single-pass harvesting operations (SPH) consist of the harvesting of logs and residues simultaneously, which is mainly associated with the whole-tree (WT) harvesting system. SPH does not involve a second extraction activity to remove logging residues but requires a central processing landing. In contrast, in double-pass harvesting operations (DPH), the primary transportation of logs and residues is performed separately and is mainly associated with the cut-to-length (CTL) harvesting system. DPH is not commonly used since its operational costs are higher than those of SPH [9]. However, if the harvest unit has a poorly maintained forest road network and processing landing, DPH can be an effective alternative for harvesting biomass. As a result, optimization efforts to enhance performance, cost-efficiency, and forest residue utilization are needed.

Several previous studies have investigated the harvesting of roundwood and energy wood using integrated harvesting systems. Most studies have included the operational performance (i.e., productivity and costs) of the harvester and forwarder [10], the feller-buncher and log-loader [7], and the cutting mechanism [11–14], as well as different logistical aspects of the wood and biomass supply chain and raw material transportation [15,16]. However, none of the past studies have provided information on the amount of logging residue left at the stump and processed at the landing when applying the SPH and DPH methods [17].

In South Korea, two integrated harvesting methods are primarily used in steep terrain (slopes exceeding 40%) to harvest merchantable and non-merchantable trees: a standard CTL system using a small shovel with a carrier [18] and a WT system using a cable yarder [19]. In cable yarding operations, WT systems have some limitations and challenges when they are used to produce high-value timber on good quality stands [20]. Expanding the practice of the WT system can reduce harvesting costs, since it can extract stems, branches, and treetops all at once [9]. This technology cuts down trees—excluding bucking, delimbing and topping at the stump—whereas processing occurs at landings that are built for that purpose [21]. This operation offers the additional advantage of higher biomass production than in stem-only harvesting methods [9,22]. However, the cable-based WT method in Korea is inherently more expensive because of the larger investment and operational costs compared to the CTL system [19,23]. In addition, the amount of logging residue left on-site after logging with these different integrated harvesting systems is largely unknown. Thus, there is a lack of data and studies with comparisons of system productivity, costs, and amount of logging residue left between the two integrated harvesting systems.

Therefore, this study's objective was to compare the operational-process time consumption and performance of two integrated harvesting methods applied to merchantable timber harvesting using a detailed time study technique. In addition, we estimated the amount of logging residue collected at the landing using both harvesting systems. It is expected that the results reported in our study can lead to informed biomass harvesting decisions and more efficient timber and biomass operations.

## 2. Materials and Methods

### 2.1. Description of the Harvest Unit and Integrated Harvesting Methods

The data in our research study was collected in forests managed by the National Forestry Cooperative Federation, located in the Korean region of Chungcheongbuk-do (37°17′27″ N/127°94′62″ E; Figure 1). The harvest unit size was 8.7 ha and the dominant species was Japanese larch (Larix kaempferi (Lamb.) Carr.), with a share of 72%. In addition, broadleaved species such as Mongolian oak (Quercus mongolica Fisch. ex Ledeb.) covered 28% of this stand. The stand density was 346 trees/ha for Japanese larch and 134 trees/ha for Mongolian oak. On average, the age of the two species was 36 years for Japanese larch and 20 years for Mongolian oak. The mean diameter at breast height (DBH) and height (Japanese larch and Mongolian oak) were 26.0 and 17.0 cm and 12.0 and 8.0 m, respectively. The mean basal area was 23.9 $m^2$/ha and the total amount of biomass (roundwood and residues) was 152.7 $m^3$/ha or 123.8 green tons (gt)/ha. The total amount of biomass was estimated from stem volume, forest biomass, and forest yield tables [24]. The harvest unit was located on a relatively steep slope (greater than 40%).

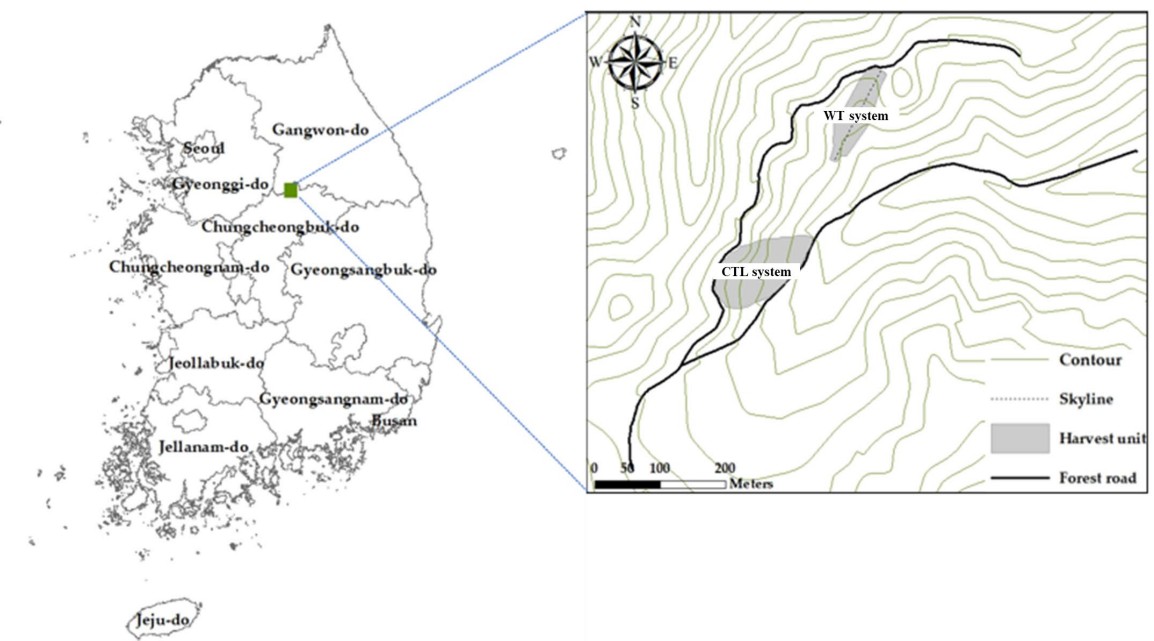

**Figure 1.** Harvesting unit in the forest of the National Forestry Cooperative Federation.

The standard CTL system consisted of two chainsaw operators for felling and processing, one small shovel for extraction from the stump to the roadside, a small-tracked excavator-mounted log grapple (5.0 tons in weight with a 0.2 $m^3$ bucket), and one tracked carrier with a small shovel (IC50L, Kato Works Co., Ltd., Shinagawa-ku, Tokyo, Japan; Figure 2) for loading and transportation from the roadside to the landing. The specific machine descriptions are presented in Table 1. Small shovels sorted and carried merchantable 2–4 m logs with logging residue (irregular circular form; Figure 2) from the stump to the roadside. The extraction activity was performed in the downhill direction. Logs and slash piles were transported to the landing by the tracked carrier for an average skid-trail distance of 252 m. A description of the harvest unit is shown in Table 2.

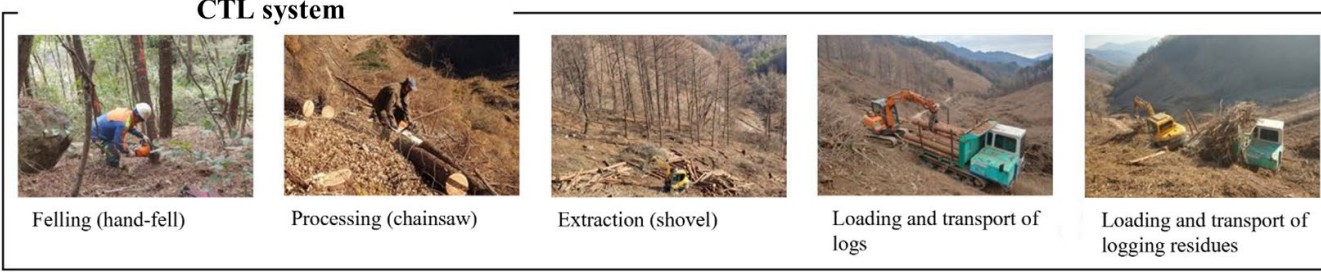

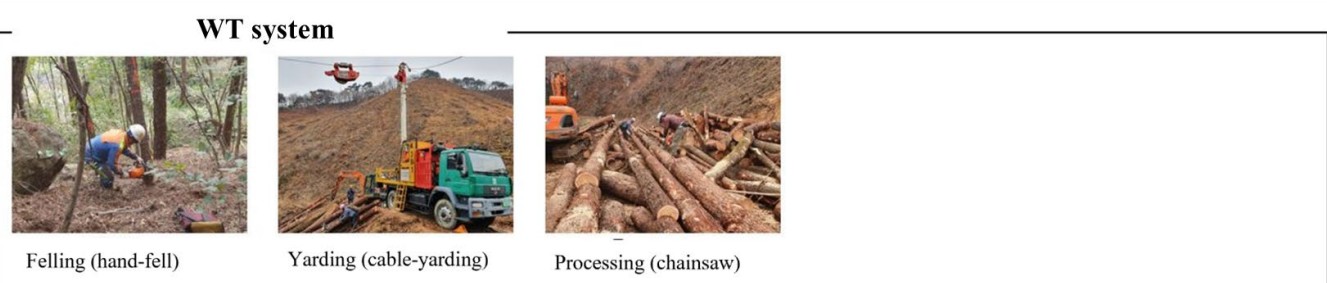

**Figure 2.** Description of the integrated harvesting systems used in the forest of the National Forestry Cooperative Federation.

**Table 1.** Specifications of the harvesting equipment used in the CTL and WT systems.

| Equipment | Company | Model | Specifications |
|---|---|---|---|
| Chainsaw | HUSQVARNA, Charlotte, NC, USA | Husqvarna 445 | Power: 2.8 hp<br>Maximum power speed: 9000 rpm<br>Fuel consumption: 980 g/h<br>Bar length: 46 cm<br>Chain type: H30<br>Gauge: 0.050"<br>Weight: 5.1 kg |
| Small shovel | Doosan, Seoul, South Korea | DX55 | Power: 56.4 hp<br>Bucket capacity: 0.2 m$^3$<br>Dimension: 1.9(W) × 1.7(L) × 2.5(H) m<br>Fuel capacity: 114 L<br>Swing speed: 9.8 rpm<br>Maximum travel speed: 4 km/h<br>Weight: 5600 kg |
| Cable yarder | Koller Forsttechnik, Schwoich, Austria | Koller K301-4 | Line pull: skyline 50 kN, mainline 26 kN<br>Line capacity<br>Skyline: 550 m, ⌀16 mm<br>Mainline: 500 m, ⌀10 mm<br>Guyline: 4 × 50 m, ⌀16 mm<br>Line speed: up to 340 m/min<br>Tower extension: 9.7 m<br>Operating range: 360°<br>Carriage payload: 160–190 kg<br>Fuel consumption: 200 L |
| Carrier | Kato Works, Tokyo, Japan | IC50L | Power: 120 hp<br>Fuel capacity: 120 L<br>Maximum payload: 4000 kg<br>Travel speed: up to 7.5 km/h<br>Dimension: 2.3(W) × 5.0(L) × 2.6(H) m<br>Weight: 6490 kg |

**Table 2.** Characteristics of the harvest unit.

| | Cut-to-Length Harvesting | Whole-Tree Harvesting |
|---|---|---|
| Area (ha) | 1.0 | 1.0 |
| Mean DBH (cm) | 25.3 | 28.6 |
| Mean height (m) | 11.7 | 12.8 |
| Basal area (m$^2$/ha) | 25.7 | 27.7 |
| Trees per ha | 523 | 450 |
| Total volume of biomass (m$^3$/ha) | 155.7 | 166.7 |
| Total volume of biomass (gt/ha) | 126.2 | 135.2 |

The WT system consisted of one chainsaw operator for tree felling, one cable yarder (Koller K301-4 truck-mounted tower yarder, Koller Forsttechnik GmbH, Kufsteiner Wald, Schwoich, Austria) for primary transportation from the stump to the landing, and one small shovel and chainsaw operator for processing and sorting logs and other waste materials at the landing (Figure 2). The cable yarder was equipped with a Koller USKA 1.5 slack pulling carriage able to hold up payloads of up to 1.5 tons. The cable extraction activity was completed by one yarder operator and two choker setters in the uphill direction. It was operated on a 40% slope and the extraction corridor had a distance of approximately 130 m.

### 2.2. Description of the Time Study Method and Forest Biomass Production Measurements

A time study technique was applied to evaluate the time consumption and productivity of the machines during the winter of 2020. Prior to felling, the trees were classified by species, measured for DBH and height, and marked with a number for an accurate evaluation of productivity. The individual machines' cycle times were observed using stopwatches to account for productive machine hours (PMH), which is separated into delays such as operational, mechanical, and personal delays. The total motor–manual felling time was classified into three main tasks: moving to a tree, preparing and clearing the workplace, and cutting down the tree. The processing activity was categorized into two main tasks (i.e., limbing and bucking) in the CTL system, whereas three tasks were included in the WT system at the landing: moving trees with the small shovel, limbing and bucking, and loading with the small shovel. Unlike moving the trees using the small shovel, the head of the machine—the grappler part—has no 360-degree rotation function. This activity phase was classified into four procedures: sorting logs and residues, throwing and pushing the logs, piling and relocating the residues, and sorting and bunching the logs and residues at the skid-trail side. Following the procedure presented by Proto et al. [25] for cable yarder machines, six work elements (outhaul, lateral out, hook-up, lateral in, in-haul, and unhook) were time-recorded in the WT system. For the CTL system, the time elements recorded for the carrier travelling from the stump to the landing included empty travel, loading by small shovel, loaded travel, and unloading time only (Table 3).

The process productivity was determined from the turn payload volume, which was calculated by measuring the large- and small-end diameters and the length of the logs. Then, log volumes were converted to oven-dried weights (oven-dried tons, odt) using a conversion ratio of 0.39 following the table of stem volume, forest biomass, and forest yield developed by the National Institute of Forest Science [24]. The cable yarding and lateral distances were measured by a TruPulse200 Laser Rangefinder (Laser Technology, Inc., Centennial, CO, USA). To estimate the logging residue collected, we employed a TXD-910F wireless axle load scale (CASKOREA CO., LTD., Seongnam-si, Gyeonggi-do, Korea; Figure 3) placed at the landing. The amount of logging residue left at the stump (hereafter the residual biomass) was estimated in both harvest units by deducting the measured weight of the collected slash piles at the landing from the estimated total biomass.

**Table 3.** The time elements measured during the time study for the cut-to-length (CTL) and whole-tree (WT) harvesting methods.

| Zone | Element | Cut-to-Length Harvesting (CTL) | | Whole-Tree Harvesting (WT) | |
|---|---|---|---|---|---|
| | | Equipment | Time Element Per Cycle | Equipment | Time Element Per Cycle |
| At stump | Felling | Chainsaw | moving to a tree, preparing and clearing the workplace, and cutting down a tree | Chainsaw | moving to a tree, preparing and clearing the workplace, and cutting down a tree |
| | Processing | Chainsaw | limbing and bucking | - | - |
| | Extraction logs and residues | Small shovel | sorting logs and residues, throwing and pushing the logs, piling and relocating the residues, and sorting and bunching the logs and residues at skid-trail side | Koller K301-4 | outhaul, lateral out, hook-up, lateral in, in-haul and unhook |
| | | Carrier | empty travel, loading by small shovel, loaded travel, and unloading at landing | | |
| At landing | Processing | - | - | Chainsaw, Small shovel | limbing and bucking |

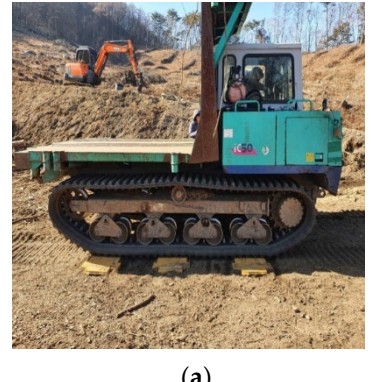
(**a**)

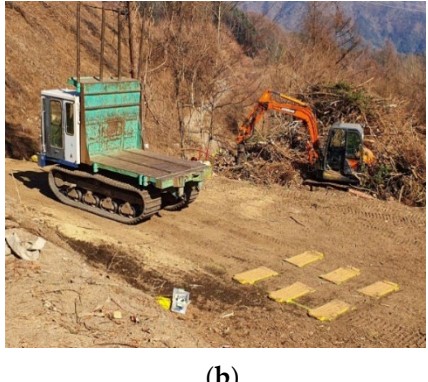
(**b**)

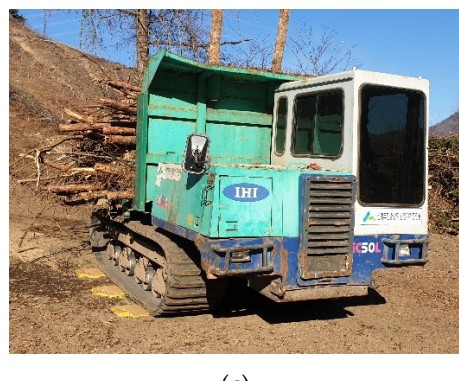
(**c**)

**Figure 3.** Overview of the logging residue weight measurement test: (**a**) measuring the empty carrier weight, (**b**) weighing the axle load scale layout, and (**c**) measuring the biomass weight with a carrier machine.

*2.3. Total System Cost Analysis*

Hourly machine costs were estimated using the traditional machine rate calculation technique, assuming 1360 scheduled machine hours (174 working days per year and 8 h shifts; [26]), as described by Miyata [27]. Fixed costs included depreciation and interest, insurance, and taxes. Depreciation was calculated based on an economic life of 7 years for all machines. The salvage value was set at 20% of the purchase price, while interest, insurance, and taxes were set at 10, 4, and 4%, respectively. Operating costs, which are defined as variable costs, were estimated based on fuel costs with a coefficient of lubrication and oil of 40% and repair and maintenance of 80%. Labor costs were set at 20.7 USD/scheduled machine hours (SMH)/persons as per the data provided by the Construction Association of Korea [28]; they included fringe benefits of 22%. According to the information on the cost elements and the hourly machine-cost evaluation, the results are presented in Table 4. Based on the above information, the total cost of the integrated harvesting system was evaluated.

**Table 4.** Cost elements and hourly machine costs for each piece of harvesting equipment.

| Cost Element | Chainsaw | Small Shovel | Carrier | Tower-Yarder |
|---|---|---|---|---|
| Sale price (USD) | 900 | 54,000 | 110,000 | 300,000 |
| Economic life (years) | 7 | 7 | 7 | 7 |
| Salvage value (%) | 20 | 20 | 20 | 20 |
| Interest rate (%) | 10 | 10 | 10 | 10 |
| Insurance and tax rate (%) | 4 | 4 | 4 | 4 |
| Fuel consumption (liter/PMH) | 1.0 | 8.0 | 7.0 | 7.0 |
| Lube and oil consumption rate (%) | 40 | 40 | 40 | 40 |
| Repair and maintenance rate (%) | 80 | 80 | 80 | 80 |
| Operator wage (USD/SMH) | 20.7 | 20.7 | 20.7 | 62.1 |
| Machine utilization rate (%) | 75 | 75 | 75 | 75 |
| Hourly machine costs (USD/PMH) | 17.5 | 39.8 | 47.9 | 113.0 |

*2.4. Statistical Analysis*

The sample size calculation could guarantee a sufficient sample size for the majority of measurable examinations. Based on Pajkoš et al. [10] and Saarillahti and Isoaho [29], the minimum sample size required was 10 at a confidence level of 1.96. We estimated the average value of the cycle time by performing a literature review and using preliminary observations for all work processing stages.

$$n = [(t^2 \times ((S \times 100)/X))/E^2]^2$$

where n = sample size, E = value of the error tolerance (i.e., 10%), t = value of the normal distribution (i.e., confidence level at 95%), S = value of the standard deviation (%), X = value of the average time consumption.

Prior to the analysis of variance (ANOVA) test, data normality and homogeneity were examined through the Shapiro–Wilk and Bartlett's tests, respectively. ANOVAs were conducted to test statistically significant differences in response variables such as DBH, height, and felling time by harvesting units (CTL vs. WT). All analyses were performed using R Statistical Software (v4.1.2; Vienna, Austria).

**3. Results and Discussion**

*Productivity and Costs of the Two Harvesting Systems (CTL and WT)*

The operating conditions, such as the DBH and height, were significantly different between each harvest unit (CTL vs. WT; ANOVA *p*-value < 0.01). The sample size varied widely among the different processes, ranging from 17 to 113 cycles.

The observed delay-free cycle time is presented in Table 5. The mean of the felling cycle time in the CTL was lower than that of the WT system due to the small size of the DBH (25.3 vs. 28.6 cm). The DBH distribution was statistically different between the two harvesting systems (ANOVA *p*-value < 0.05). Among the cycle elements of the felling activity, the moving time from one tree to another was the highest in both the CTL and WT systems, at 43% and 37%, respectively. The processing cycle time took approximately two-times longer for the WT system than for the CTL system, primarily because of the remarkable increase in the bucking time caused by the chainsaw operators avoiding risks from the small shovel at the landing. In CTL, the extraction of logs and residues was performed separately, taking a substantially higher time than the WT system, where whole trees were moved at once to the landing in each cycle. Therefore, the total extraction time including for logs and residues in CTL was 13.9 min/odt—approximately 100% higher compared to that of WT (6.9 min/odt). The average yarding distance and the lateral yarding distance were 70.2 m and 8.5 m on average, respectively, and the average yarding distance in CTL was 70 m. The extraction in the WT system consumed the highest amount of time in the hook-up element cycle due to the interference of the spread of logging slash on the ground from operators' movements. The transportation cycle time from the roadside

to the landing by one-tracked carrier only occurred in the CTL system because in the WT system, the trees were extracted from the stump to the landing directly by the cable yarder. The loaded travel time for log transportation was about 32% higher than that of residue transportation because the average payload by truck was considerably higher than that of residue. In the case of the CTL system, mean cycle times were 22.0 ± 1.4 min for 'transporting logs' and 17.0 ± 0.8 min for 'transporting residues'.

**Table 5.** Mean cycle time at each operational activity in both CTL and WT.

| Cycle Elements | CTL System | | | | WT System | | |
| --- | --- | --- | --- | --- | --- | --- | --- |
| | Sec/Cycle | | | % | Sec/Cycle | | % |
| | Mean | SD [a] | | | Mean | SD [a] | |
| Felling | 82.0 | ±94.5 | | 100 | 88.2 | ±56.1 | 100 |
| Moving | 35.0 | ±59.9 | | 42.7 | 32.6 | ±37.4 | 37.0 |
| Clearing workplace | 24.2 | ±51.7 | | 29.5 | 25.4 | ±35.9 | 28.8 |
| Cutting down | 22.8 | ±13.2 | | 27.8 | 30.2 | ±18.8 | 34.2 |
| Processing | 89.0 | ±70.2 | | 100 | 184.7 | ±45.6 | 100 |
| Limbing | 65.4 | ±59.9 | | 73.4 | Recorded as processing time | | |
| Bucking | 23.6 | ±16.7 | | 26.6 | | | |
| Extraction | 1311.3 | - | | 100 | 325.3 | ±74.3 | 100 |
| Shovel logging (Log) | 734.7 | - | | 100 | | N/A | |
| Shovel logging (Residue) | 576.6 | - | | 100 | | N/A | |
| Out-haul | | | | | 30.0 | ±11.8 | 9.7 |
| Lateral out | | N/A | | | 72.2 | ±39.1 | 21.9 |
| Hook-up | | | | | 85.1 | ±44.0 | 25.8 |
| Lateral in | | | | | 50.3 | ±22.7 | 15.3 |
| In-haul | | | | | 42.5 | ±19.9 | 13.6 |
| Unhook | | | | | 45.2 | ±28.6 | 13.7 |

| Cycle Elements | Log | Residue | Log | Residue | Log | Residue | |
| --- | --- | --- | --- | --- | --- | --- | --- |
| Transportation | 1315.6 | 994.1 | ±353.2 | ±184.9 | 100 | 100 | |
| Empty travel | 201.4 | 155.1 | ±69.8 | ±44.7 | 15.3 | 15.6 | N/A |
| Loading | 794.4 | 577.4 | ±253.8 | ±134.4 | 60.4 | 58.1 | |
| Loaded travel | 300.2 | 247.6 | ±83.0 | ±45.9 | 22.8 | 24.9 | |
| Landing | 19.6 | 14.0 | ±2.9 | ±3.9 | 1.5 | 1.4 | |

[a]: Standard deviation.

The productivity and costs for the integrated harvesting were observed for each work processing stage separately (Table 6). The productivity was 1.45 odt/PMH at a cost of 86.8 USD/odt for the conventional CTL system, while the production rate for the WT system was 2.99 odt/PMH at a cost of 45.5 USD/odt. In this case study, the WT system's productivity was 100% higher than that of the CTL system.

The performance of the individual working activities differed slightly between the two harvesting systems. Felling time in the WT system was 19% higher than in the CTL system, primarily due to the larger turn size. The average turn payload during extraction was considerably different between the two systems, with the cable-based extraction having payloads 20% lower than the small shovel. The processing productivity of the WT system was twice higher than that of CTL due to the larger turn size and safer work conditions. The CTL system delivered 9.41 odt/PMH of logs and 1.45 odt/PMH of residue. The total volume of logging residue collected at the landing for the CTL and the WT systems was approximately 0.5 and 2.1 odt/ha, respectively (Figure 4). After the collection of residues by the small shovel in the CTL system, 16.1 odt/ha of logging residue (including branches and foliage) was left at the stump as a residual. As for the WT system, the residual was 16.5 odt/ha (approximately 35% of the total amount of logging residues). Furthermore, we found that an average of 26% of the total forest biomass vanished before or during

harvesting and transport. As a result, the amount of logging residue left at stumps was higher for the CTL than for the WT system.

**Table 6.** Mean delay-free cycle time by harvesting systems.

| Configuration | Mean Cycle Time (Sec/Cycle) | Turn Size (Odt/Cycle) | Productivity (Odt/PMH) | Cost (USD/Odt) |
|---|---|---|---|---|
| CTL system | | | | |
| Felling | 82.0 | 0.13 | 5.71 | 3.1 |
| Processing at stump | 89.0 | 0.13 | 5.17 | 3.4 |
| Extraction | 1311.2 | 1.37 | 3.75 | 10.6 |
| Transporting logs | 1315.6 | 3.44 | 9.41 | 9.3 |
| Transporting residues | 994.1 | 0.40 | 1.45 | 60.4 |
| System total | - | - | 1.45 | 86.8 |
| WT system | | | | |
| Felling | 88.2 | 0.17 | 6.77 | 2.6 |
| Extraction | 325.3 | 0.27 | 2.99 | 37.8 |
| Processing on landing | 184.7 | 0.57 | 11.29 | 5.1 |
| System total | - | - | 2.99 | 45.5 |

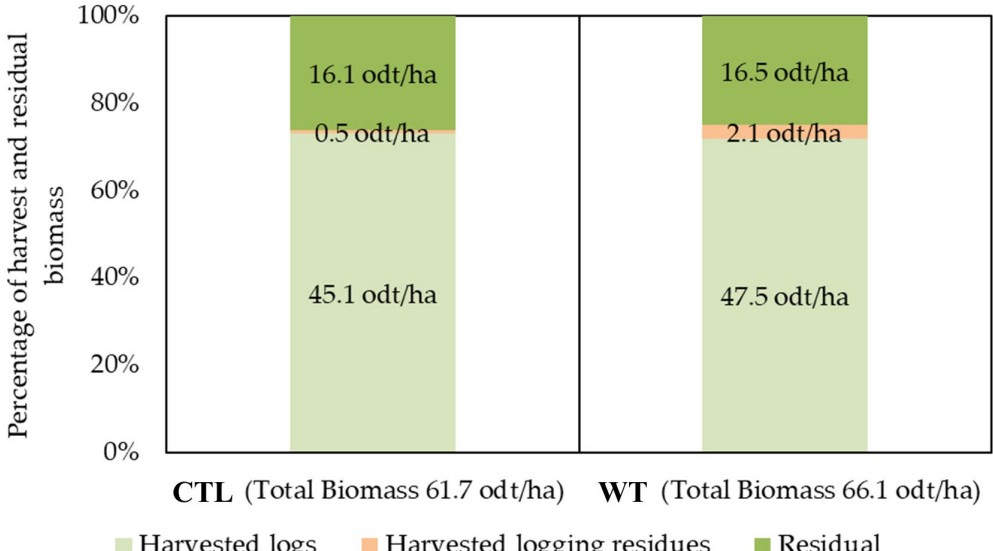

**Figure 4.** The amount of logging residue left and collected for each harvesting system.

In biomass recovery operations, the WT system is the preferred method compared to CTL. For example, Briedis et al. [17] and Han et al. [21] reported that the WT system was an efficient and economical method when logs and residues are transported simultaneously. In those studies, the cost of the WT system was 20%–50% lower than that of the CTL systems—particularly in thinning [30] and clear-cutting [21] operations. The SPH method reduces harvesting costs compared to DPH as there is no need to extract and transport logging residues separately [7]. Our analysis indicated that the WT system was less expensive for more than 45% of the study area. In the CTL system, processing costs increased because the machines had to move between standing trees and processed one tree at a time within the harvest unit [21,31,32]. In contrast, our results showed that WT had higher costs than CTL due to use of integrated processing operations (combined chainsaw operator and small shovel).

With the WT system, a higher biomass volume was produced at the landing compared to CTL. For example, the logging residues left ranged from 20 to 90 odt/ha, depending on the harvesting method [9,22]. In WT, the amount of recovered biomass on the sites was 10%–50% of the total volume of harvested materials such as softwood, hardwood,

and mixed stands [17,21,33]. In CTL, the amount of logging residue left ranged from 57 and 97% at the stump [9]. Our results imply that WT can reduce the amount logging residue, since a lower leakage was associated with processing and transporting (included extraction; [17,21,34]). In addition, after WT, the logging residues removed on the landing had larger-diameter residues than CTL, consistent with the results in Hytönen and Moilanen [17]. Furthermore, according to Finnish silvicultural guidelines, after harvesting, recommendations suggest leaving 30% of logging residues on the site due to the adequate nutrients left for reforestation and for the growth of the remaining trees [29] as cited in [8]. In our tests, more logging residue was left on the harvesting unit after both harvesting than in the guidelines. As a result, the amount of logging residue left at the harvest unit was associated with the harvesting methods.

The application of the WT system and its technology may be more cost-effective than the CTL system when both logs and logging residue must be supplied to bioenergy and bioproducts industries. In contrast, a stem-only harvesting system could be a more cost-effective harvesting technology than in the case of the WT system when there are no markets for biomass, and due to the higher extraction and transport costs involved [21,22]. Thus, the WT system seems to be a much more efficient option when additional logging residue extraction and transporting costs are involved—as in the case of the CTL system.

The productivity, costs, and the amount of logging residue removed of two integrated harvesting methods—a single-pass harvesting operation (WT) and a double-pass harvesting operation (CTL)—were investigated in this study. In the WT system, a cable yarder extracted whole trees to the landing after the chainsaw-felling operation, followed by tree processing performed with a chainsaw and a small shovel. In the CTL system, the small shovel and carrier brought logs and logging residues to the roadside separately after the chainsaw felling and the processing operation at the stump. Differences in performance were observed between the two harvesting systems. Future research should include the performance of biomass conversion technologies, such as chipping, grinding, and bundling, and a comparison of logging residue quality (i.e., size uniformity and contamination from dirt) between the two harvesting systems.

### 4. Conclusions

The combined harvesting of logs and residues is the most common biomass harvesting technique worldwide and is used to increase biomass recovery efficiency, particularly in North America and Europe. Two integrated harvesting methods have been used in Korea. The CTL system uses a chainsaw to fell and process trees at the stump, bringing logs and residues by a small shovel and carrier separately from the stump to the landing, while the WT system transports full trees to the landing where they are processed to produce commercial logs and residues. Due to the lack of comparative studies on system performance and harvesting residue yield between the CTL and WT systems, this study aimed to evaluate the performance of the two systems using a detailed, practical time study method and to estimate the amount of logging residue harvested on the landing. The system production rate for the WT system was approximately twice-higher than that of the CTL system, with noticeably lower costs. In addition, more residues were brought to the landing with the WT system (2.1 odt/ha), as well as material of a larger diameter than that of the CTL system (0.5 odt/ha). Furthermore, after CTL and WT, the residual biomass was 25% of the total amount of biomass in each harvest unit. The percentage of residual biomass value should be considered with its potential variance, since the total biomass of each harvest unit was estimated in this study.

**Author Contributions:** Conceptualization, H.W., E.L. and S.-K.H.; methodology, H.W. and E.L.; validation, H.W. and S.-K.H.; formal analysis, H.W., M.A., H.C. and E.L.; investigation, E.L. and S.-K.H.; resources, E.L. and S.-K.H.; data curation, H.W. and E.L.; writing—original draft preparation, H.W., M.A. and E.L.; writing—review and editing, H.W., M.A. and S.-K.H.; visualization, H.W., E.L. and S.-K.H.; supervision, M.A. and S.-K.H.; project administration, E.L. and S.-K.H.; funding acquisition, S.-K.H. All authors have read and agreed to the published version of the manuscript.

**Funding:** This research was funded by the Korea Forest Service (Korea Forestry Promotion Institute), grant number 2021351B10-2223-AC03.

**Data Availability Statement:** Not applicable.

**Conflicts of Interest:** The authors declare no conflict of interest.

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
