# Peer review of "The Impact of Integrated Harvesting Systems on Productivity, Costs, and Amount of Logging Residue in the Clear-Cutting of a Larix kaempferi (Lamb.) Carr. Stand"

_forests, doi:10.3390/f13111941_

Round 1
Reviewer 1 Report
This is a very good article, it raises important issues. The methods used are not a very modern approach, rather commonly known, but applied in the conditions and environment that have not been studied so far. The article provides new results and conclusions in line with the current state of knowledge.
Detailed comments:
The title reflects the content of the article well.
The abstract is very well written.
Keywords are well written.
Introduction.
"Various integrated harvesting systems have been applied to implement biomass harvesting technologies (e.g., singlepass and double-pass operations) under a wide range of harvest unit conditions, such as slope and road surface, and landing size [5, 7]."
The authors can expand the analysis in the sentence above with the article "Warguła, Ł., Kukla, M., Krawiec, P., & Wieczorek, B. (2020). Impact of number of operators and distance to branch piles on woodchipper operation. Forests, 11 (5), 598. " which analyzes the impact of the number of machine operators on the collection of biomass during the chipping of branches.
In paragraphs 68 to 74, authors may add literature describing the analysis of energy wood harvesting due to:
- cutting mechanism:
Spinelli, R., Cavallo, E., Eliasson, L., Facello, A., & Magagnotti, N. (2015). The effect of drum design on chipper performance. Renewable Energy, 81, 57-61.
Warguła, Ł., Kukla, M., Wieczorek, B., & Krawiec, P. (2022). Energy consumption of the wood size reduction processes with employment of a low-power machines with various cutting mechanisms. Renewable Energy, 181, 630-639.
Nati, C., Eliasson, L., & Spinelli, R. (2014). Effect of chipper type, biomass type and blade wear on productivity, fuel consumption and product quality. Croatian Journal of Forest Engineering: Journal for Theory and Application of Forestry Engineering, 35 (1), 1-7.
Spinelli, R., Cavallo, E., Eliasson, L., & Facello, A. (2013). Comparing the efficiency of drum and disc chippers. Silva Fennica, 47 (2).
- raw material transport
Gałęzia, T. (2013). Energy balance and time-consumption of selected components in technological chain of forest biomass harvesting. Sylwan, 157 (6), 419-424.
Material and methods
The description of devices from line118 should indicate the power of the drive units of the devices used, eg power of petrol saws, excavators, etc. Such device characteristics are commonly used. Additionally, it allows for more detailed analyzes. The list of machines could be in a table with their characteristics, giving the power of the machines is very important.
Resalts
The results are presented correctly.
I believe that the section Discussion and conclusions should be separated for the readability of the article. Additionally, in Forests such an arrangement is recommended.
Line 267-270, the authors should refer the figures presented there to the results of other researchers' research.
References
About 40% of the literature is older than five years, which I believe authors can easily correct. The problem of auto-citation is not recognized in the literature.
The article is interesting and important for the industry. It presents an analysis of known methods in an environment and conditions not previously studied.
Author Response
Nov 1st 2022
Dear editor,
Thanks for your review and suggestions for this manuscript. My co-author and I have revised previous manuscript to response your comments and suggestions. Please find enclosed a revised copy of our manuscript.
The attached pages summarize how we have addressed the reviewers’ suggestions and comments. The comments and suggestions provided by the reviewers were helpful in improving the technical presentation of our study results and the readability of the manuscript and we are grateful about that. We believe that our paper become much more clear and informational, compared to the previous version of the manuscript.
Thank you,
Corresponding author

Reviewer 2 Report
Suggestions and comments to the article
1. Change the keywords so that they do not overlap with the title of the work
2. It would be valuable to show the time structure of individual machines, e.g. by giving the percentage of measured activities
3. Characterise in more detail ALL machinery and equipment used in the process of harvesting and transporting wood (biomass), as the characteristics of the machinery directly affect its performance
4. I suggest providing information on what was the average lateral distance of transport by cable yarder within WT system and average distance of transport by shovel in the CTL system
5. I suggest adding what was the total operating time of the various machines and how much biomass was harvested during the study?
6. The use of ANOVA analysis is unnecessary when determining differences between two samples (CTL vs. WT). A Student's t-test would have been better. Proving that trees with greater BHD and height were harvested in the WT system does not add much to the work anyway - it is another factor that differentiates the technologies.
7. I am not convinced that testing for differences in felling time on the two surfaces was necessary. I suggest removing the information about this testing from the methodology. In general, the statistical analyses performed can be disregarded, possibly - and this would be valuable - to try to construct models of efficiency or time-consumption
8. For an accurate characterization of the work process, Table 4 is worth supplementing with information on cycle time variation, turn size variation, productivity variation and costs variation - e.g. by giving minimum and maximum values. I also suggest adding a column here with information on the number of cycles measured
9. The analyses of the amount of logging residue (Figure 4) should be approached with caution, as the total biomass was estimated. Since it was not measured, it is worth commenting on it in the paper
10. In the discussion, I suggest including comparisons of the technologies analyzed to standard technologies commonly used in Korean forests
Author Response

(The authors gave the same response as above.)

Reviewer 3 Report
Review
Achieving an increase in the intensity of the use of forest biomass, especially through the utilization of logging residues for energy purposes, has been a characteristic goal of most forestry-developed countries in the world for a long time. These efforts have recently intensified in connection with the lack of other, especially fossil, energy sources and efforts to increase the share of renewable energy sources, to which logging waste undoubtedly belongs. It can therefore be concluded that the presented article deals with an issue that is current on a global scale and brings new knowledge about it in the conditions of the Republic of Korea. The article can therefore be considered beneficial for forestry science and forestry practice, not only in the Republic of Korea, but in principle its topic is also interesting for interested experts from other countries.
Chapter 1 - Introduction first characterizes the consumption and structure of energy sources in South Korea and highlights the increasing efforts to use renewable energy sources and reduce greenhouse gas emissions to zero by 2050. In this chapter, the important concept of "integrated harvesting" is also explained in a comprehensible form, discussed in the article, and the application of integrated logging in the conditions of South Korea is characterized. When preparing Chapter 1, the authors appropriately worked with the information sources cited in the References section at the end of the article. Here, the authors also clearly formulate the goal of their research activities, the results of which form the basis of the presented article.
Chapter 2 – Materials and Methods firstly in sub-chapter 2.1 concisely characterizes the localities where the research investigations were carried out, including the characteristics of the structure of the forest stands. In addition, the technical means and technological procedures for solving both types of harvesting, which were examined and evaluated by the authors, are described here. Subchapter 2.2 describes the method of time measurement studies and the method of measuring biomass production. This description is sufficiently comprehensible and the methods used are factually correct in my opinion. The comprehensibility of the description is effectively supplemented by table 2. Here I ask: did the authors check the validity of the "conversion ratio of 0.39" for their chosen research locations, or were they just taken over apparently general data from the source [18] (line 166)? I would recommend that in Fig. 3, the photo on the right side should be replaced by a photo of the weighing machine with harvesting waste loaded, this way there are only two photos showing the principle of weighing the empty machine. Subchapter 2.3 provides data for cost analysis. Is Table No. 3 data taken over or found directly by the authors during research investigations? Subchapter 2.4 explains the principle of statistical evaluation of results. It is rather short, but acceptable.
Chapter 3 – Results First, in subchapter 3.1, it presents the results of determining the productivity and costs of both monitored mining systems. The authors state here the high significance of the results of the monitored mining methods (line 212), which is certainly gratifying. Table 4 shows the main results achieved. I have a question about this: these results are given as "hard" numbers, given to two decimal places. However, they are apparently determined as average values from individual partial measurements or calculations. In my opinion, the values of standard deviations from the arithmetic mean of individual quantities should also be listed here. I have no comments on the remaining part of subsection 3.1. I have substantive comments on subchapter 3.2: I recommend changing the name of the subchapter, which is very similar to the name of subchapter 3.1, but the text of subchapter 3.2 deals with the amount and structure of individual categories of biomass. Similarly, as I mentioned in the comment to subsection 3.1, I recommend justifying how it is possible that the values reported here have no variance.
Chapter 4 – Discussion and conclusions appropriately comments on the achieved results and compares them with the results of other authors. In my opinion, the results achieved are supported by this comparison. Although the scope of the author's collective measurements is not very large and is related to only two localities, the achieved results can be accepted at least at an indicative level. It would certainly be desirable to continue the research and arrive at findings that have more general validity. Even in this form, however, the results are beneficial and I recommend publishing the article after slight adjustments.
Comments formal:
- in the Abstract, unusual units are used to express the amount of harvesting waste "odt", e.g. on line 26 - I recommend first stating the word designation of the unit, i.e. "oven dried tonne" and only then using the abbreviation "odt". In other parts of the text of the article, this principle of indicating the meaning of unit abbreviations is continuously used
- I have a similar opinion on the use of the PMH mark - it is also not a common unit, such as ha, kg, m, etc., I recommend first stating the word designation of the unit and only then using the abbreviation PMH
- on line 41, the space between the words "by2050" is missing
- on line 120, the exponent in the expression 0.2 m3 – correctly 0.2 m3.
Author Response

(The authors gave the same response as above.)

Reviewer 4 Report
Brief summary:
The manuscript presents an interesting study that examines the impact of integrated harvesting systems on productivity, costs and amount of logging residues in clear-cut forest management. The study is relevant for steep terrains where whole tree harvesting is quite common and, in many cases, includes the logging residues utilisation. The finding is derivated from a single location located in Korea. The authors build results on cost calculations and basic statistical analysis.
General concept comments
There is a lack of novelty in the studied harvesting system. The topic is relevant, but the authors should have to pay more attention to the investigation of existing literature. CTL and WT are broadly used in northern America and Scandic countries. The main problem with the rationale is that the manuscript is motivated to increase the use of renewable biomass to enhance the transition towards a climate-neutral economy. Still, it does not present the role of residues in wood-based energy. It briefly mentions the merchantable and non-merchantable trees in the introduction section, but it is not clear what is the management preferences in the study (e.g. production of energy wood). Are also logs used for energy purposes or only logging residues? Furthermore, the materials and method provide most of the information needed to ensure the reproducibility of the study. Some minor improvements are possible (for example, in the cases of describing (i) the cost calculation method and (ii) the definition of logging residues vs. residues).
However, a major backbone of the manuscript is the lack of complexity of the method. The methodology is very limited. I would expect a more detailed analysis. The current methodology is not adequate for the scope and metric of MDPI: Forest Journal. Authors should pay attention to the methodology and presentation of results in other studies published in MDPI journals. For example:
- Spinelli, R.; Magagnotti, N.; Cosola, G.; Engler, B.; Leitner, S.; Vidoni, R. Fuel and Time Consumption in Alpine Cable Yarder Operations. Forests 2022, 13, 1394. https://doi.org/10.3390/f13091394
- Latterini, F.; Venanzi, R.; Stefanoni, W.; Sperandio, G.; Suardi, A.; Civitarese, V.; Picchio, R. Work Productivity, Costs and Environmental Impacts of Two Thinning Methods in Italian Beech High Forests. Sustainability 2022, 14, 11414. https://doi.org/10.3390/ su141811414
I see many options for going into more complexity e.g.:
- investigating the studied harvesting systems in more detail with additional statistical tests of detailed processes.
- going into more detailed economic evaluation by valuing also the revenue from harvested and residual biomass, taking into consideration the supply chain costs. It would be nice for the reader to understand if there is any benefit in such forest management. If not, explain why there is interest in such activities?
- the study is motivated by a zero emissions policy it would be proper to investigate the environmental impact of the studied harvesting system (GHG emissions analysis... )
- A similar suggestion on environmental impacts is to further investigate nutrient extraction differences in the case of studied harvesting systems.
The results section lacks both tables and figures. For example, table 3, which presents the harvesting equipment costs, could be presented with much more detail (e.g maintenance, operator wage, overhead and profit, totals also in SHM). A similar suggestion goes to Table 4: System total or mean time consumption of the felling cycle is missing, although it is mentioned in the text (lines 223-224).
The author's present manuscript added value in estimating the amount of logging residues collected at the landing using both harvesting systems. The difference between »harvested logging residues« and »residual« is unclear. I suggest introducing the difference in the materials section to ensure the reproducibility of the study. For example, Figure 4 mentions »Harvested logging residues« while the paragraph above mentioned all residues (harvested added by residual) remained at the stump (e.g Figure 4 or at lines 244-249).
Specific comments:
- Consistently check the aberration throughout the manuscript (e.g. CTH, CT or CTL, same with WT, WTH..).
- Check references (e.g. 20,22 are not reproducible).
- The manuscript considers US $; I would suggest authors present also a conversion ratio of US $ to EU € (from the period of conversions). To gain comparability with EU studies and ensure further citations.
Some minor comments:
Line 9: Author List and Affiliations: Authors' full first and last names must be provided. The initials of any middle names can be added. The PubMed/MEDLINE standard format is used for affiliations: complete address information including city, zip code, state/province, and country.
Line 41: add space »by2050«
Line 95: remove special characters (-)
Line 161: Change CTH to CTL
Line 209: Inadequate citation of the source of the software (https://ropensci.org/blog/2021/11/16/how-to-cite-r-and-r-packages/ )
Line 221: Transporting logs and residue information are marked with »– for no information«. In my opinion, this is included in Extraction. As presented now, it is confusing the reader.
Line 225: Language editing suggested.
Line 233-234: Add decimal places to mean cycle times.
Line 240-241: "In latter case" is it referring + to the CTL system?
Line 326-328: Remove the template version of acknowledgements. If there is no one authors wish to acknowledge, this section may be deleted.
Author Response

(The authors gave the same response as above.)

Round 2
Reviewer 1 Report
All comments were taken into account, corrected and explained.
Author Response
Dear Reviewer,
Again, thanks for your review and suggestions for this manuscript. The comments and suggestions provided by the reviewers were helpful in improving the technical presentation of our study results and the readability of the manuscript and we are grateful about that. We believe that our paper become much more clear and informational, compared to the previous version of the manuscript.
Best regards,
Reviewer 4 Report
Thanks to the authors of the manuscript for the revisions. There are improvements, but I feel the authors have not provided the expected clarifications on major comments that I pointed out. As a reviewer, I appreciated that the authors responded point by point, at least to minor remarks and highlighted the changes in the article.
Major comments:
In this second review, the major comments remain the same as in the first review. Even though the authors have added a table with average times at each activity, I still feel there is much potential to improve the methodology. I pointed out some of the ideas already in the first review. I still see some critical aspects in the "materials and methods" section, and the authors could further improve the introduction/discussion.
Some other comments:
- The main problem with the rationale is that the manuscript is motivated to increase the use of renewable biomass to enhance the transition towards a climate-neutral economy. Still, it does not present the role of residues in wood-based energy.
- It briefly mentions the merchantable and non-merchantable trees in the introduction section, but it is not clear what is the management preferences in the study (e.g. production of energy wood or merchantable trees).
- The authors responded that CTH was revised to CTL but did not consider the figures.
- The author's present manuscript added value in estimating the amount of logging residues collected at the landing using both harvesting systems. However, the difference between »harvested logging residues« and »residual« remains unclear. It could be just a terminology issue. Also, the authors shall point out the total biomass was estimated, not measured, and it, therefore, needs to be considered with caution.
- Improvements in methodological sections and results (e.g. cost variations) are needed to explain calculations of costs.
- The authors remain inconsistent. Tables 5 and 6 present information in one case, referred to as average time, and in another case, referred to as mean. The numbers (if doubled) should have at least the same number of significant decimal places.
- Table 5 presents "Recorded as processing time" in WT, but on the other hand, authors can calculate percentages in the following column. Explain how?
Author Response
Dear Reviewer,
Again, thanks for your review and suggestions for this manuscript. The comments and suggestions provided by the reviewers were helpful in improving the technical presentation of our study results and the readability of the manuscript, and we are grateful for that. We believe that our paper has become much more clear and informational, compared to the previous version of the manuscript.
Best regards,
